# Dimensionality reduction of longitudinal 'omics data using modern tensor factorizations

Uria Mor[1,2�], Yotam Cohen[1☍], Rafael Valdés-Mas[1☍], Denise Kviatcovsky[1], Eran Elinav[1,3‡], Haim Avron[2‡]*

**1** Systems Immunology Department, Weizmann Institute of Science, Rehovot, Israel, **2** School of Mathematical Sciences, Tel Aviv University, Tel Aviv, Israel, **3** Microbiome and Cancer Division, DKFZ, Heidelberg, Germany

☍ These authors contributed equally to this work.
‡ EE and HA also contributed equally to this work.
* haimav@tauex.tau.ac.il

**Data Availability Statement:** All relevant data are within the manuscript and its Supporting information files. Source code for analysis is

## Abstract

Longitudinal 'omics analytical methods are extensively used in the evolving field of precision medicine, by enabling 'big data' recording and high-resolution interpretation of complex datasets, driven by individual variations in response to perturbations such as disease pathogenesis, medical treatment or changes in lifestyle. However, inherent technical limitations in biomedical studies often result in the generation of feature-rich and sample-limited datasets. Analyzing such data using conventional modalities often proves to be challenging since the repeated, high-dimensional measurements overload the outlook with inconsequential variations that must be filtered from the data in order to find the true, biologically relevant signal. Tensor methods for the analysis and meaningful representation of multiway data may prove useful to the biological research community by their advertised ability to tackle this challenge. In this study, we present ᴛᴄᴀᴍ—a new unsupervised tensor factorization method for the analysis of multiway data. Building on top of cutting-edge developments in the field of tensor-tensor algebra, we characterize the unique mathematical properties of our method, namely, 1) preservation of geometric and statistical traits of the data, which enable uncovering information beyond the inter-individual variation that often takes over the focus, especially in human studies. 2) Natural and straightforward out-of-sample extension, making ᴛᴄᴀᴍ amenable for integration in machine learning workflows. A series of re-analyses of real-world, human experimental datasets showcase these theoretical properties, while providing empirical confirmation of ᴛᴄᴀᴍ's utility in the analysis of longitudinal 'omics data.

## Author summary

Tensor methods have proven useful for exploration of high-dimensional, multiway data that is produced in longitudinal 'omics studies. However, even the most recent applications of these methods to 'omics data are based on the canonical polyadic tensor-rank

available on GitHub: https://github.com/UriaMorP/tcam_analysis_notebooks/.

**Funding:** This research was partially supported by the Israeli Council for Higher Education (CHE) via the Weizmann Data Science Research Center. The funders had no role in study design, data collection and analysis, decision to publish, or preparation of the manuscript.

factorization whose results heavily depend on the choice of target rank, lack any guarantee for optimal approximation, and do not allow for out-of-sample extension in a straightforward manner. In this paper, we present a method for tensor component analysis for the analysis of longitudinal 'omics data, built on top of cutting-edge developments in the field of tensor-tensor algebra. We show that our method, in contrast to existing tensor-methods, enjoys provable optimal properties on the distortion and variance in the embedding space, enabling direct and meaningful interpretation, supporting traditional multivariate statistical analysis to be performed in the embedding space. Due to the method's construction using tensor-tensor products, the procedure of mapping a point to the embedding space of a pre-trained factorization is simple and scalable, giving rise to the application of our method as a feature engineering step in standard machine learning workflows.

This is a *PLOS Computational Biology* Methods paper.

## Introduction

Recent developments in high-throughput methodologies enable the assessment of molecular entities from biological samples on a global scale at steadily decreasing costs, allowing to conduct biological and clinical studies at previously unfeasible magnitude, in terms of the number of biological repetitions and molecules quantified [1]. A consequence of the increased availability of 'omics methods is the possibility to conduct large-scale longitudinal studies prospectively following participants in a variety of clinical contexts. In particular, longitudinal 'omics profiling, combined with clinical measurements, enable to detect and understand individual changes from baseline, improving personalized health and medicine by using tailored therapies [2].

Yet, despite the surge of longitudinal multiomics studies, the toolset for analyses which fully utilize multiway structures in the data remains limited to date, with only a handful of applicable algorithms and software being suitable for specific tasks [3–5].

Higher-order tensors (multiway arrays of numbers) are arguably the most natural data structures for describing high-dimensional, multiway data such as longitudinal 'omics data. Indeed, an impressive adoption of tensor factorization methods for time-series analysis has recently emerged, allowing trajectory analysis for microbiome data [6–8] as well as neural dynamics [9]. Generally referred to as tensor component analysis (TCA) [9], these multiway dimensionality reduction methods for 'omics data provide a view on global—multivariate variations in the data, thus complementing methods such as [3–5] for univariate time-series analysis. The TCA is most often computed using CANDECOMP/PARAFAC (CP) factorization [10, 11], which dramatically limits the ability to apply machine learning (ML) algorithms, as it does not allow for straightforward mapping of unseen data points to the reduced space. In addition, CP-based TCA requires choosing the number of components (dimensions) to be considered, since different choices may result in significantly different transformations of the data, additional uncertainties in analyzing complex information are introduced. Moreover, when alternating least squares (ALS) is used for finding the CP factors, there is no guarantee that the resulting factors will correspond to the best low-rank approximation.

In this article, we present TCAM, a method for unsupervised dimensionality reduction which provides an answer to the unmet need for trajectory analysis of longitudinal 'omics data (Fig 1). Our method is based on a cutting-edge mathematical framework (the **M**-product between tensors), which allows for a natural generalization of the notion of singular value

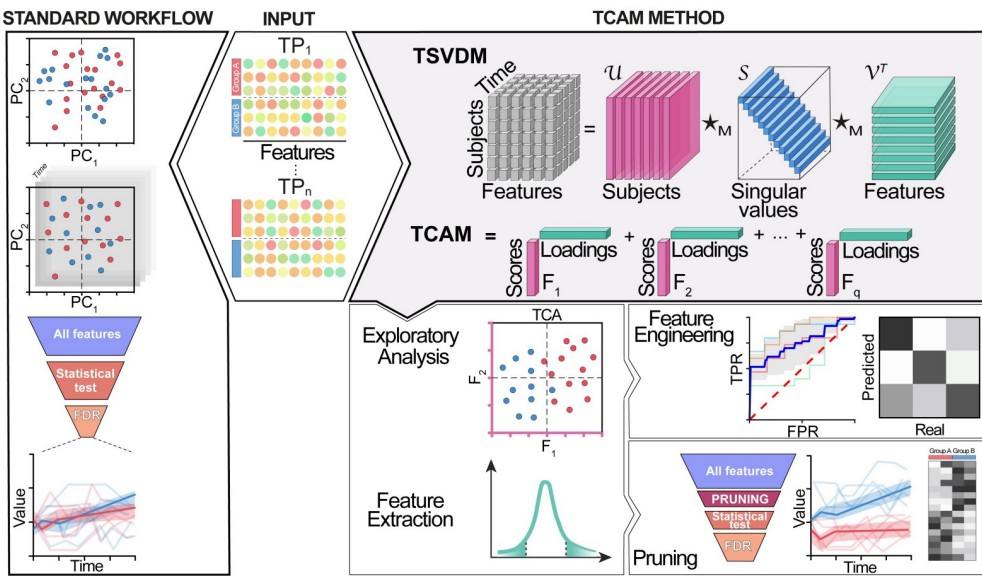

**Fig 1. Illustration of settings and workflow.** Center rhombus describes the typical data produced in a longitudinal experiment, where high-dimensional samples are collected from multiple subjects across various timepoints. Left-hand side describes a summarized overview of standard, non-tensor-based workflows, including (top to bottom) ordination plots with repeated measurements, per-timepoint multivariate analysis, and funnels for discovery using univariate time-series analysis. Top right—schematic derivation of TCAM from TSVDM. Bottom right—TCAM's output and its applications, including exploratory analysis of the data through a reduced features space where variation between points reflects differences between high-dimensional temporal trajectories, feature engineering for downstream ML workflows, and feature selection for downstream univariate exploration.

decomposition (SVD) for matrices ($2^{nd}$ order tensors) to higher order tensors [12]. In contrast to existing ALS-CP-based methods, TCAM factors do not depend on prior choice of target rank, resulting in consistent outputs between executions. Additionally, given a fitted TCAM factorization, the embedding of a new trajectory is straightforward, unlike CP-based methods in which such computation is untrivial. Moreover, TCAM enjoys provable optimality regarding the variance and distortion of the embeddings. Finally, we stress that unlike other tensor methods (e.g., [6]), that rely on statistical properties of the data and its generative process, TCAM makes no assumptions regarding the structure of the data, making it suitable for any choice of normalization method. Thus, allowing the usage of TCAM with diverse data types (e.g., different 'omics sources) and providing the flexibility needed to answer different kinds of questions.

Following the formal construction of TCAM, we present a series of comparative analyses, using real-world data collected from longitudinal 'omics studies. These experiments, and their TCAM analyses, highlight how TCAM overcomes both the shortcomings of using traditional, matrix-based workflows for the analysis of these data, and those that remain when employing existing, state-of-the-art tensor methods. Given its theoretical guarantees, TCAM should be applicable to any kind of 'omics data (assuming the adequate normalization of the data). Empirical evidence for this claim is given through the re-analysis of a longitudinal proteomics dataset, in which we utilized TCAM to uncover insights that were not mentioned in the original work from which the data was taken (though established in the literature). Finally, we showcase the straightforward application of TCAM to supervised machine learning workflows, where TCAM serves as a 'drop-in' feature-engineering step in the pipeline by the virtue of its out-of-sample extension.

## Models

### Preliminaries and notations

A real tensor of order-$N$, denoted by $\mathcal{A} \in \mathbb{R}^{d_1 \times d_2 \times \cdots \times d_N}$, is a multi-dimensional array with real entries indexed by $N$-tuples. For example, the $i_1, i_2, \ldots, i_N$ entry of $\mathcal{A}$ is denoted by $\mathcal{A}_{i_1, i_2, \ldots, i_N}$. In this paper, we consider $3^{rd}$ order tensors $\mathcal{A} \in \mathbb{R}^{m \times p \times n}$ holding data from $p$-dimensional samples, collected from $m$ subjects across $n$ time-points. The size of $p$ is determined by the number of features measured in the 'omics method being used, e.g., it can be the number of observed bacterial species in metagenomics sequencing or the number of genes in transcriptomics. We consider a "subject centered" view of the data tensor, that is, viewing the order-3 tensor as an $m$-long "list of matrices" where each matrix corresponds to a time-series collected from a single individual, as illustrated in Fig 2a and 2b. We use MATLAB notations for slicing and indexing of tensors, e.g., $\mathcal{A}_{i,:,:} \in \mathbb{R}^{1 \times p \times n}$ denotes the $i^{th}$ *horizontal slice* of $\mathcal{A}$, which may be considered as a $p \times n$ matrix. *Fibers* of $\mathcal{A}$ are obtained by fixing two indices. Of particular interest are the *tube-fibers* that are the $n$ dimensional vectors $\mathcal{A}_{i,j,:}$.

We briefly define the tensor-tensor product following the construction in [12]. Let $\mathbf{M}$ be an $n \times n$ invertible matrix, then the mode-3 product the tensor $\mathcal{A}$ with the matrix $\mathbf{M}$ is denoted by the tensor $\widehat{\mathcal{A}} = \mathcal{A} \times_3 \mathbf{M}$ whose tube-fibers are given by $\widehat{\mathcal{A}}_{i,j,:} = \mathbf{M}\mathcal{A}_{i,j,:}$. Note that since $\mathbf{M}$ is non-singular, it holds that $\widehat{\mathcal{A}} \times_3 \mathbf{M}^{-1} = \mathcal{A}$. The face-wise product of tensors $\mathcal{A} \in \mathbb{R}^{m \times p \times n}$ and $\mathcal{B} \in \mathbb{R}^{p \times r \times n}$ is denoted by the tensor $\mathcal{C} := \mathcal{A} \triangle \mathcal{B} \in \mathbb{R}^{m \times r \times n}$ where each slice $\mathcal{C}_{:,:,i}$ is given by the matrix product $\mathcal{A}_{:,:,i}\mathcal{B}_{:,:,i} \in \mathbb{R}^{m \times r}$. The $\star_{\mathbf{M}}$ tensor-tensor product of $\mathcal{A}$ and $\mathcal{B}$ is defined as $\mathcal{C} := (\widehat{\mathcal{A}} \triangle \widehat{\mathcal{B}}) \times_3 \mathbf{M}^{-1}$. Given a non-singular $n \times n$ matrix $\mathbf{M}$, the tubal singular value decomposition with respect to the $\star_{\mathbf{M}}$-product (TSVDM) of $\mathcal{A}$ is written as $\mathcal{A} = \mathcal{U} \star_{\mathbf{M}} \mathcal{S} \star_{\mathbf{M}} \mathcal{V}^{\mathbf{T}}$ where $\mathcal{U}, \mathcal{V}$ are $\star_{\mathbf{M}}$-orthogonal tensors and $\mathcal{S}$ is f-diagonal (Fig 3) (see S1 Text for formal definitions).

Kilmer et al. have recently established an Eckart-Young like—best low-rank approximation results for the case where $\mathbf{M}$ is a non-zero multiple of an orthogonal matrix [12]. Briefly, the best t-rank $q$ (multi-rank $\rho$) approximations of a tensor $\mathcal{A}$ are obtained by t-rank $q$ (multi-rank $\rho$) truncation of $\mathcal{A}$'s TSVDM. In this work, we further generalize these results to a novel notion of explicit rank truncation (see S1 Text). To maintain consistency, throughout all

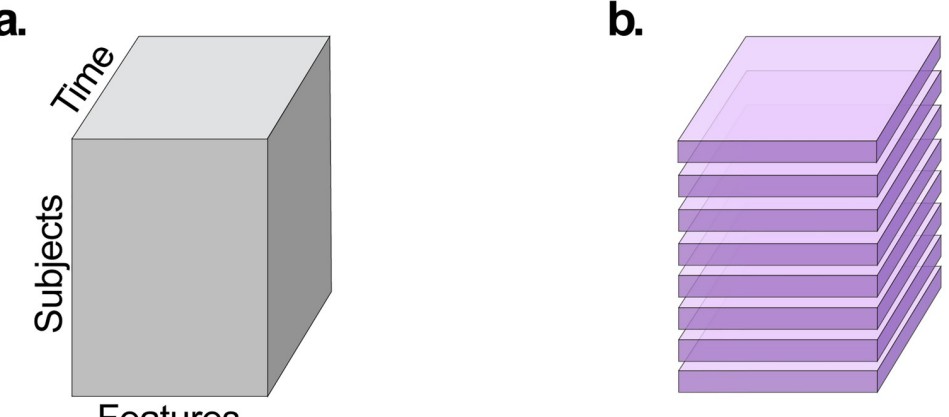

**Fig 2. Subject centered view of $3^{rd}$ order tensor. a** An illustration of the data structure. **b** The right panel presents a breakdown of the left tensor into $m$ horizontal slices that are $p \times n$ matrices.

**Fig 3. Illustration of the TSVDM decomposition for a $3^{rd}$ order tensor.** Left hand side of the equation shows that data tensor $\mathcal{A}$, right hand side shows the factors $\mathcal{U} \in \mathbb{R}^{m \times m \times n}$, $\mathcal{S} \in \mathbb{R}^{m \times p \times n}$, $\mathcal{V}^{\mathrm{T}} \in \mathbb{R}^{p \times p \times n}$, where $\mathcal{U}$, $\mathcal{V}$ are $\star_{\mathbf{M}}$-orthogonal tensors and $\mathcal{S}$ is f-diagonal.

demonstrations in this study, we considered $\mathbf{M}$ defined by the (scaled) Discrete Cosine Transform (DCT-II). However, we encourage TCAM users to experiment with different transforms, e.g. the software implementation allows for sampling a random $\mathbf{M} \in \mathrm{O}(n)$ at uniform (Haar) distribution.

## Proposed method: TCAM

The mean sample of a tensor $\mathcal{A}$, is defined as $\bar{\mathcal{A}} = {}^1/_m \sum_{i=1}^{m} \mathcal{A}_{i,:,:} \in \mathbb{R}^{1 \times p \times n}$. A tensor $\mathcal{A}$ is in **mean-deviation form** (MDF), if $\| \bar{\mathcal{A}} \|_F^2 = 0$, where $\|\cdot\|_F$ denotes the Frobenius norm: $\| \mathcal{A} \|_F^2 = \sum_{k,j,i} \mathcal{A}_{k,j,i}^2$. Any tensor $\mathcal{A}$ can be centered to MDF by subtracting its mean sample from each horizontal slice. So, henceforth we assume that $\mathcal{A}$ is in MDF.

Given a tensor $\mathcal{A}$ in MDF, with a factorization $\mathcal{A} = \mathcal{U} \star_{\mathbf{M}} \mathcal{S} \star_{\mathbf{M}} \mathcal{V}^{\mathrm{T}}$, the ranking vector $\mathbf{r}$ of $\mathcal{A}$ is an ordered set of tuples $\mathbf{r} = \{(\mathbf{r}_{h,1}, \mathbf{r}_{h,2})\}_{h=1}^{pn}$ where $\mathbf{r}_{h,1} \in \{1, \ldots, p\}$ and $\mathbf{r}_{h,2} \in \{1, \ldots, n\}$ denote indexes of diagonal entries of $\widehat{\mathcal{S}}$ such that

$$\widehat{\mathcal{S}}_{\mathbf{r}_{1,1}, \mathbf{r}_{1,1}, \mathbf{r}_{1,2}} \geq \widehat{\mathcal{S}}_{\mathbf{r}_{2,1}, \mathbf{r}_{2,1}, \mathbf{r}_{2,2}} \geq \cdots \geq \widehat{\mathcal{S}}_{\mathbf{r}_{np,1}, \mathbf{r}_{np,1}, \mathbf{r}_{np,2}} \tag{1}$$

The TCAM of $\mathcal{A}$ is defined by a *scores* matrix $\mathbf{Z} \in \mathbb{R}^{m \times pn}$ whose entries are

$$\mathbf{Z}_{\ell,h} = [(\mathcal{A} \star_{\mathbf{M}} \mathcal{V}) \times_3 \mathbf{M}]_{\ell, \mathbf{r}_{h,1}, \mathbf{r}_{h,2}} \tag{2}$$

and an $np \times p$ *loadings* matrix $\mathbf{V}$ with entries $\mathbf{V}_{h,j} = \widehat{\mathcal{V}}_{\mathbf{r}_{h,1} j, \mathbf{r}_{h,2}}$. **Algorithm 1** for obtaining TCAM highlights that most of the computational effort is due to the TSVDM, which requires $\mathcal{O}(n(pm^2 + m^3))$ arithmetic operations, omitting the overhead involved with applying the $\times_3$ $\mathbf{M}$ operations that are of negligent cost compared to that of the SVD computation. The explicit rank-$q$ truncated TCAM is obtained by keeping the first $q$ columns and rows of $\mathbf{Z}$ and $\mathbf{V}$ respectively. Each row of the factors matrix represents the $p$-dimensional time-series (trajectory) of each subject, while the loadings matrix measures the contribution—magnitude and direction —of each of the $p$ 'omics features to each of the TCAM factors across samples.

Given a time-series of samples $\mathcal{X} \in \mathbb{R}^{1 \times p \times n}$, not necessarily included in the horizontal slices of $\mathcal{A}$, the transformation of $\mathcal{X}$ to the reduced dimensional space defined by the TCAM fitted using $\mathcal{A}$ is given by **Algorithm 2** and is equal to

$$\mathbf{x}_h = [(\mathcal{X} \star_{\mathbf{M}} \mathcal{V}) \times_3 \mathbf{M}]_{\ell, \mathbf{r}_{h,1}, \mathbf{r}_{h,2}} . \tag{3}$$

This ability to transform new data points to the reduced features space makes TCAM amenable for use as a feature-engineering step in supervised ML workflows.

**Algorithm 1** TCAM construction
**Input:** Data $\mathcal{A} \in \mathbb{R}^{m \times p \times n}$
**Parameters:** Orthogonal matrix $\mathbf{M} \in \mathbb{R}^{n \times n}$
1: $\bar{\mathbf{A}} \leftarrow (\sum_{i=1}^{m} \mathcal{A}_{i,:,:})/m$
2: $\mathcal{U}, \mathcal{S}, \mathcal{V} \leftarrow \text{TSVDM}(\mathcal{A} - \bar{\mathbf{A}})$
3: $\mathbf{r}_h \leftarrow (r_{h,1}, r_{h,2})$ (see Eq 1), $\widehat{\sigma}_h \leftarrow \widehat{\mathcal{S}}_{r_{h,1}, r_{h,1}, r_{h,2}}$
**Output:**
  Explained variance portion: $\{\widehat{\sigma}_h^2/(\sum_{j=1}^{np} \widehat{\sigma}_j^2)\}_{h=1}^{np}$
  Transformation parameters: $\mathbf{r}, \mathcal{V}, \bar{\mathbf{A}}$

**Algorithm 2** TCAM projection
**Input:** $\mathcal{X} \in \mathbb{R}^{1 \times p \times n}$ a single (multivariate) time-series
**Parameters:** TCAM transform parameters: $\bar{\mathbf{A}}, \mathbf{r}, \mathcal{V}$
1: $\mathcal{Z} \leftarrow (\mathcal{X} - \bar{\mathbf{A}}) \star_{\mathbf{M}} \mathcal{V}$
2: $\widehat{\mathcal{Z}} \leftarrow \mathcal{Z} \times_3 \mathbf{M} \in \mathbb{R}^{1 \times p \times n}$
3: $\mathbf{z}_h \leftarrow [\widehat{\mathcal{Z}}]_{1, r_{h,1}, r_{h,2}}$
**Output:** $\mathbf{z} \in \mathbb{R}^{np}$

The transformation in Eq 3 (Fig 4), is a member of the family of pseudo $\star_{\mathbf{M}}$-orthogonal, explicit rank-$q$ truncated tensor-to-vector mappings (see S1 Text). As such, it enjoys two fundamental properties: variance maximization and minimization of the distortion. Formally, let $\mathbf{Z} = Q(\mathcal{A}) \in \mathbb{R}^{m \times q}$ denote the image of a tensor $\mathcal{A} \in \mathbb{R}^{m \times p \times n}$ (in MDF) under a pseudo $\star_{\mathbf{M}}$-orthogonal explicit rank-$q$ truncated tensor-to-vector mapping $Q$, then

1. The sample-variance of the image, $\text{Tr}(\mathbf{Z}^{\mathbf{T}}\mathbf{Z})/(m-1)$, is maximized when $Q$ is defined by the explicit rank-$q$ truncated TCAM.

2. The distortion in the configuration caused by $Q$, $\|\mathbf{A}\mathbf{A}^{\mathbf{T}} - \mathbf{Z}\mathbf{Z}^{\mathbf{T}}\|_*$, where $\mathbf{A} \in \mathbb{R}^{m \times np}$ is the matrix obtained by mode-1 unfolding of $\mathcal{A}$, is minimized when $Q$ is defined by the explicit rank-$q$ truncated TCAM.

Combined, these properties provide a formal justification for applying multivariate hypothesis testing methods such as PERMANOVA [13] to the reduced space representation of the data, in addition to meaningful and intuitive interpretation for the results of such methods. Proofs for these statements, along with additional details, are given in S1 Text.

We now remark regarding the choice of $q$. The dimension of the reduced space depends on the purpose of the analysis. For example, one might choose $q$ based on the traditional

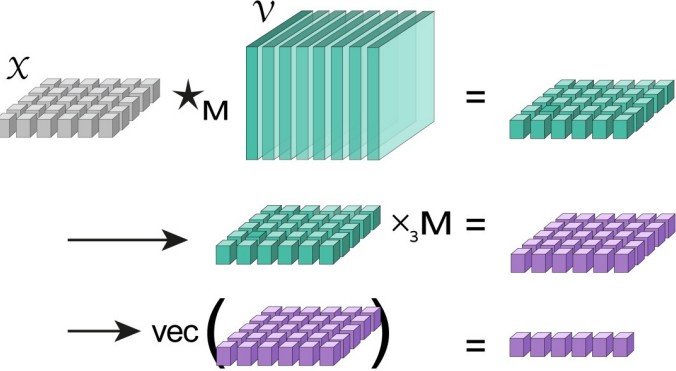

**Fig 4. Illustration of the TCAM mapping defined in Eq 3 and Algorithm 2.** Top: right multiplication of new data point $\mathcal{X} \in \mathbb{R}^{1 \times p \times n}$ (a matrix) by $\mathcal{V}$, followed by application of $\mathbf{M}$ (middle), and concatenation (bottom).

considerations for choosing the number of principal components, such as signal—noise ratio assumptions or based on scree plots. Regardless of the considerations for choosing $q$, it should be noted that unlike CP, where the target rank of the approximation affects the resulting embedding, the TCAM factors are pre-determined by the data (and **M**), up to a variance and distortion invariant multiplication by a unitary tensor. Thus, the configuration reflected by the first $i$ TCAM components remains unchanged when taking any $q \geq i$.

## Results

### TCAM reveals information beyond inter-individual variations

In this first experiment, we wish to exemplify the shortcomings of traditional, matrix-based, dimensionality reduction methods when applied to longitudinal 'omics data. More often than not, longitudinal 'omics data is characterized by high inter-individual differences, regardless of time or state in which samples were taken. Traditional matrix-based dimensionality reduction methods such as PCA (which attempt to find a representation in a reduced dimensional space in which the variance is maximized) are prone to mask interesting temporal variations by highlighting the prominent inter-individual differences. To demonstrate this point, we use data from a work by Suez et. al. [14], investigating the reconstitution of the gut microbiome in healthy individuals following antibiotics administration. In the original study, participants were split into three study arms—21 day-long probiotics supplementation (PBX), autologous fecal microbiome transplantation (aFMT) derived from a pre-antibiotics treated sample, or spontaneous recovery (CTR). Stool samples were collected at baseline (BAS, days 0 to 6), during antibiotics treatment (ABX, days 7 to 13) and the intervention phase (INT, days 14 to 42).

Indeed, PCA of all time-points resulted in a representation that mainly reflects inter-individual differences (Fig 5a and S1(a) Fig), while temporal intra-individual variations remained obscured. The high correlation between "baseline configuration" and "complete configuration" (Pearson, $\rho = 0.72$, $p < 10^{-10}$, Fig 5b) implies that the repeated samples add very little information to the ordination. Similarly, a per-phase perspective of the data did not capture changes in microbiome composition, but a mere snapshot of temporal trends (S1(b), S1(c), S1 (d) and S1(e) Fig). These results demonstrate that even in the presence of a temporal perturbations as substantial as the effect of antibiotics treatment on the microbiome, PCA is unable to utilize the longitudinal sampling for picking up signals, which were attainable in the case of single timepoint study design. In contrast, application of TCAM to the data following log-folds baseline normalization (LFB, S1 Text), generated a coherent representation of the data (Fig 5c) where each point in the reduced space represents a complete temporal trajectory of a subject throughout the entire experiment. Additionally, the TCAM scores approximate the true distances between input trajectories (S1 Text) providing clear and accurate interpretation of the configuration in the resulting embedding and proportion of variation explained by each ordination axis. Multivariate hypothesis testing revealed significant differences between trajectories of the FMT group and those of PBX (Fig 5c; PERMANOVA, $P < 0.05$). In light of the normalization scheme used, these result renders that courses of change in gut microbiome composition of subjects supplemented with probiotics following antibiotics administration, significantly differed from those of individuals who underwent autologous FMT—in agreement with the findings of Suez et. al.

To further investigate the sources of variation between the groups, we considered the highest magnitude TCAM loadings associated with $F_1$ (Fig 5d), which accounts for $\sim 11\%$ of the variation in the data by itself, and exhibits significant differences between groups (ANOVA, $P < 0.05$). Inspection of the top 5% contributing features to the variation in $F_1$, highlighted five probiotic species of large-positive magnitude, namely *B.breve, B.bifidum, L.acidophilus, L.*

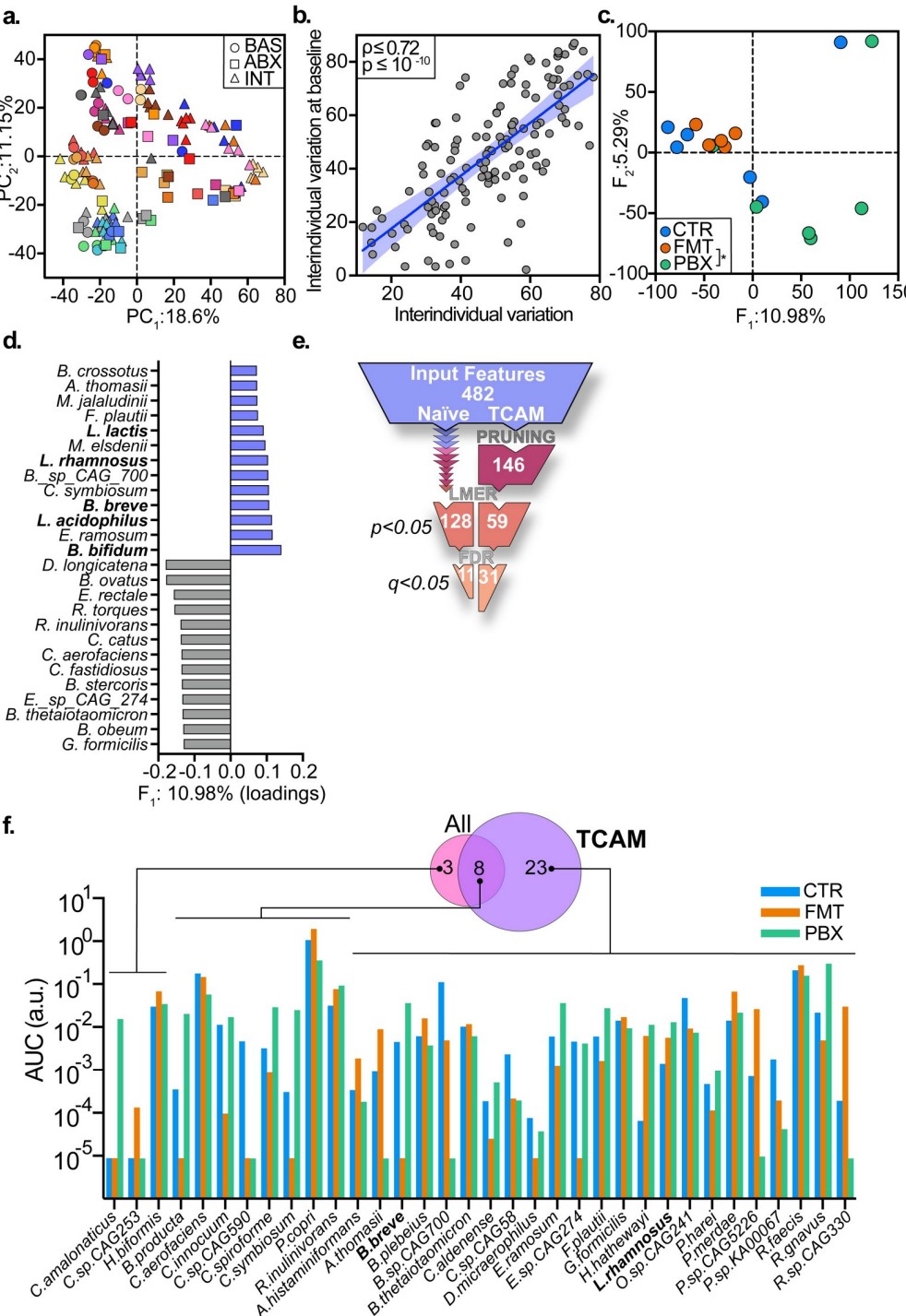

**Fig 5. Comparison of TCAM with existing matrix-based methods. a** PCA of all timepoints, colored by participant. **b** Regression line of mean distance between subjects at all timepoints (x) and at baseline (y). Distances computed using $PC_1$ and $PC_2$. **c** Leading TCAM factors. **d** Bar graph showing top 2.5% features contributing to $F_1$s variation. **e** Comparison of discovery rates for univariate hypothesis testing (lmer), between naïve workflow (left) and TCAM-based pruning (right) workflow. **f** Venn diagram and bar graphs. Bars denote per-subject iAUC for all detected bacteria (q<0.05). Venn diagram relates each bacterium to the workflow it was detected in. Bars represent medians. Names of microbial species which are included in the probiotics mix are highlighted in bold.

*rhamnosus* and *L.lactis*, meaning that higher values of these species in the LFB normalized data are "pushing" $F_1$ scores upwards (or, to the right-hand side of Fig 5c). Thus, we conclude that these probiotic species which were in fact consumed by the PBX group [14], may play a major role in separating the PBX group from the rest of the cohort, providing a strong demonstration that results obtained using TCAM have meaningful biological interpretation.

Routine exploration for features exhibiting differential trends between groups across time typically involves univariate time-series hypothesis testing such as repeated measures ANOVA and linear mixed effect model (lmer), followed by false discovery rate correction necessitated by the number of features being tested. Considering the typically large number of features in 'omics data, combined with the assumption that only a few of these features actually differ between the groups, this strategy may prove too stringent, discarding true signals due to an arbitrary large initial number of features. Indeed, applying lmer to each species in the initial dataset (482 features), resulted in 128 taxa showing different temporal trends between groups, reflected by the statistically significant estimation of the interaction term (S1 Text), with only 11 bacteria maintaining statistically significant values after correction for multiple testing, none of which is a probiotic species (S2(a) and S2(b) Fig). In contrast, when TCAM was used as a pruning strategy, by considering only top TCAM loadings (S1 Text) to reduce the number of initial features being tested, the final set of species presenting significantly different temporal trends between groups contained twenty-three new bacteria (two of these are probiotic species, Fig 5e and 5f and S3 Fig), and an overlap of eight species discovered with and without pruning (S2(a) Fig). The TCAM-based pruning strategy failed in the detection of three species, that would otherwise have been discovered (S2(b) Fig).

To demonstrate the application of TCAM to the exploratory analysis of dense longitudinal time-series, we used the ECAM dataset [15], which contains stool microbiome collected in high frequency during the first two years of life for 43 infants. In this example, we put our focus on the well characterized differences between development course of microbiomes of infants that were vaginally delivered (V), and those of infants delivered by cesarean section (C). Observing the leading TCAM factors of the data, we notice significant differences between microbiome trajectories of the two modes, highlighted both by univariate test and a (linear kernel) support vector machine classifier (SVC) (S4(a) Fig; PERMANOVA; $P < 0.01$). To find the bacteria of highest contribution to the separation between the temporal trajectories of the two modes of birth, we computed the magnitude of the feature loadings when projected onto the normal direction to the boundary of decision (S4(b) Fig). Indeed, among the top 1% contributing features, we were able to identify bacteria such as *Bacteroides*, *Bifidobacterium* and *Enterobacteriaceae* that were previously found to exhibit differential behavior across time between the birth modes. Applying univariate tests following the same pruning strategy as in the post-antibiotics reconstitution example, resulted in a fine-grained, coherent view of the discriminatory features (S4(c) and S4(d) Fig).

## Comparison with existing tensor-based methods

Next, we evaluated TCAM's performance in comparison to a state-of-the-art tensor factorization method: Context-aware Tensor Factorization (CTF) [6]. Unlike TCAM, CTF is designed specifically for 16S metagenomic sequencing data, and substantially utilizes the compositionality and sparsity of the data by finding a CP decomposition which best approximates the non-zero values. For this demonstration, we chose the resistant starch type 4 (RS4) interventional dataset, comparing the effect of four different types of RS4 fiber administration (tapioca, maize, corn and potato) on the microbiome composition measured using 16S sequencing of stool samples collected each week during a five weeks long trial [16]. The four arms of this experiment were

defined by the source of fibers: tapioca and maize groups represent sources of fermentable fibers, while potato and corn groups mostly contain fibers that are inaccessible for microbiome degradation, thus they are considered control groups. In the original paper, the authors noticed significant differences in specific time-points in the tapioca and maize groups, but did not report any results of time-series analysis.

CTF was applied to the count data while considering five components (S1 Text), resulting in significant differences noted between trajectories of participants in the maize group from those of individuals supplemented with corn and potato (PERMANOVA; $P < 0.05$, Fig 6a), with significantly different factor scores between the groups obtained for the fourth CTF component (accounting for 0.11% of the squared sum of singular values, ANOVA; $P = 0.005$). However, when considering the features of highest contribution to variability of the scores on the fourth factor for downstream—univariate time-series analysis, none of the selected features have shown a significantly different temporal pattern between groups. In contrast to CTF, in which the factorization is inseparable from the robust centered log-ratio (rclr) normalization scheme, TCAM makes very few assumptions regarding the data and its generative model, thus allowing higher degree of flexibility when normalizing the data. First, on top of (matrix) rclr normalization [17] (S1 Text), we further employed deviation from baseline transformation (DFB, S1 Text) before applying TCAM to the data, in order to have a view of the data that is centered at each individual's deviation from their personal baseline. Indeed, truncated TCAM resulted in significant differences between maize group to the remaining study groups, in addition to observed differences between the corn and tapioca groups (PERMANOVA; S5(a) Fig). Leading TCAM loadings associated with the factors differentiating between groups were considered for time-series analysis (S1 Text), which highlighted the differences in the relative abundance levels of *P.distasonis* and *Enterobacteriaceae* between the groups (lmer; $P < 0.05$, $Q < 0.1$, S5(b) and S5(c) Fig). To further demonstrate TCAM's flexibility, we applied the factorization to the same dataset following LFB normalization (S1 Text). The alternative normalization uncovered significantly different temporal trends between the maize group to all of the remaining groups in the cohort (PERMANOVA; $P < 0.05$, Fig 6b), and the pruning strategy revealed seven bacteria, including the above mentioned taxa, featuring a statistically significant trend throughout time (lmer; $P < 0.05$, $Q < 0.05$, Fig 6c and 6d S5(c) Fig). Moreover, using the top loadings associated with $F_3$, we highlighted additional features demonstrating patterns of increasing bacteria in the form of *Lachnospiraceae* in the maize group and *P. distasonis* in the tapioca group (Fig 6e).

## Universal applicability to 'omics data

To assess TCAM's applicability to 'omics data other than metagenomics, we use the proteomics data set from Sailani et. al. [18] concerning seasonal patterns of the human microbiome, transcriptome, metabolome and proteome. The original study cohort contains data collected from 105 individuals, where each participant donated about twelve samples during a three-year period (one sample every three months). Here, we set the focus on a subset of the data which contains proteomics samples of individuals featuring insulin sensitivity (IS) or insulin resistance (IR), and addressed the differences between proteomic trajectories of these two groups throughout time.

Using TCAM following DFB normalization of the data (see S1 Text), we detected a significant separation between IR and IS groups based on first factor's scores (t-test; $P < 0.05$), suggesting that a considerable portion of the data's variability is explained by differences between these groups (Fig 7a). Similar to our analysis framework above, we turned to the top loadings associated with the first factor (Fig 7b). Among the top ranked proteins, we could easily notice

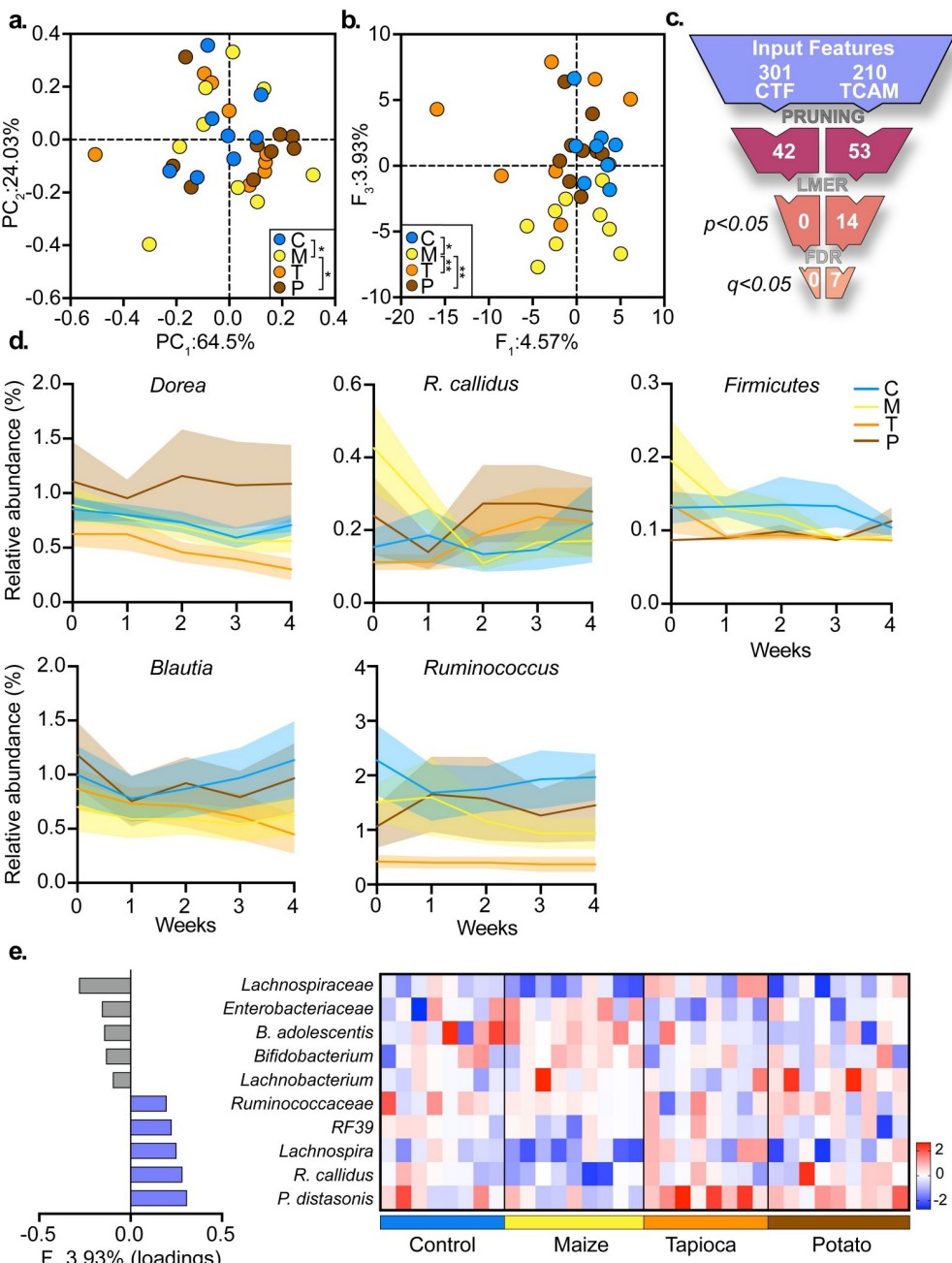

**Fig 6. Comparison of TCAM with existing tensor-based methods. a** Scatter plot of the data from [16] obtained using CTF; Inset: pairwise PERMANOVA. **b** TCAM Scatter plot for data of [16] using TCAM; Inset: Pairwise PERMANOVA **c** Funnel comparing discovery rates of CTF and TCAM based pruning strategies. **d** Time series describing significant bacteria (lmer) found using TCAM based pruning strategy on top of LFB normalization. **e** Barplot with top and bottom 2.5% loadings for $F_3$. Heatmap representing per-subject AUC (log scale) for the same features; Color bar indicates z-score normalized value.

angiotensinogen (AGT), which was previously associated with IS condition [19]; paraoxonase-1 (PON1), that has been found to down-regulate insulin resistance in mice [20]; apolipoprotein-3 (APOC3), highly associated with IR [21]; and increasing $\alpha_2$-HS-glycoprotein (AHSG), which is indeed tightly associated with IS [22]. The high level of consistency between these

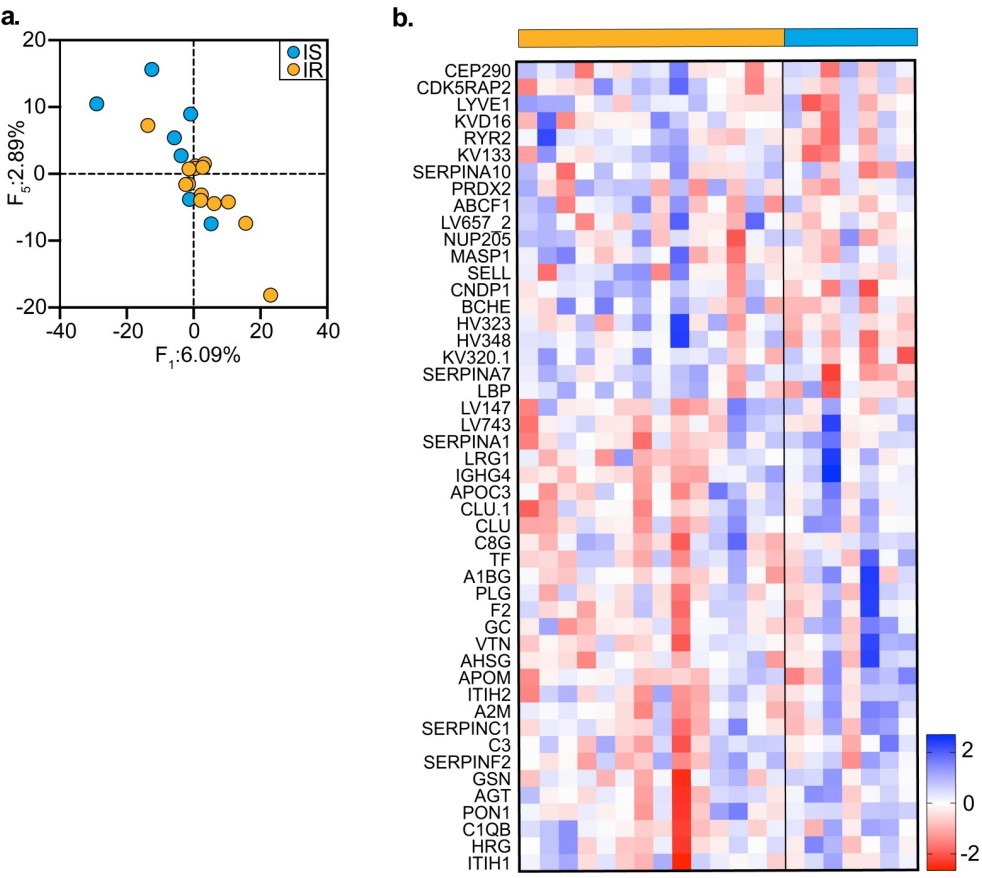

**Fig 7. TCAM's applicability to proteomics datasets. a** Scatterplot of leading TCAM factors significantly correlated with insulin resistance or sensitivity. Points are colored according to insulin resistant (IR) and insulin sensitive (IS) information. **b** Heatmap showing the sum of top and bottom 25 features contributing to the variation on $F_1$ according to their loadings; Color bar indicates z-score normalized value.

TCAM-produced findings and the existing literature convinces that our method is generally applicable to 'omics data.

## Application to supervised ML

In the context of standard supervised-ML, classification or regression models are trained using a labeled training set, and are expected to generally apply to (unlabeled) inputs. Note that TCAM is essentially an unsupervised dimensionality reduction method, unlike label aware methods such as Avocado [8], which takes into account phenotypic information for computation of the embedding. Thanks to its natural out-of-sample extension (Eq 3), TCAM is amenable for seamless integration as a feature engineering step in supervised-ML workflows. To demonstrate this ability, we utilized a 16S rRNA metagenomics dataset from a study by Schirmer et al., which contains stool samples collected from pediatric UC patients monitored for 52 weeks under three different treatments. The original study characterized microbial dynamics along disease course, in light of host response to each of the applied treatments [23]. Here, we constructed a supervised ML model which uses longitudinal metagenomics data to classify disease status labeled by flare (FLR) and remission (REM).

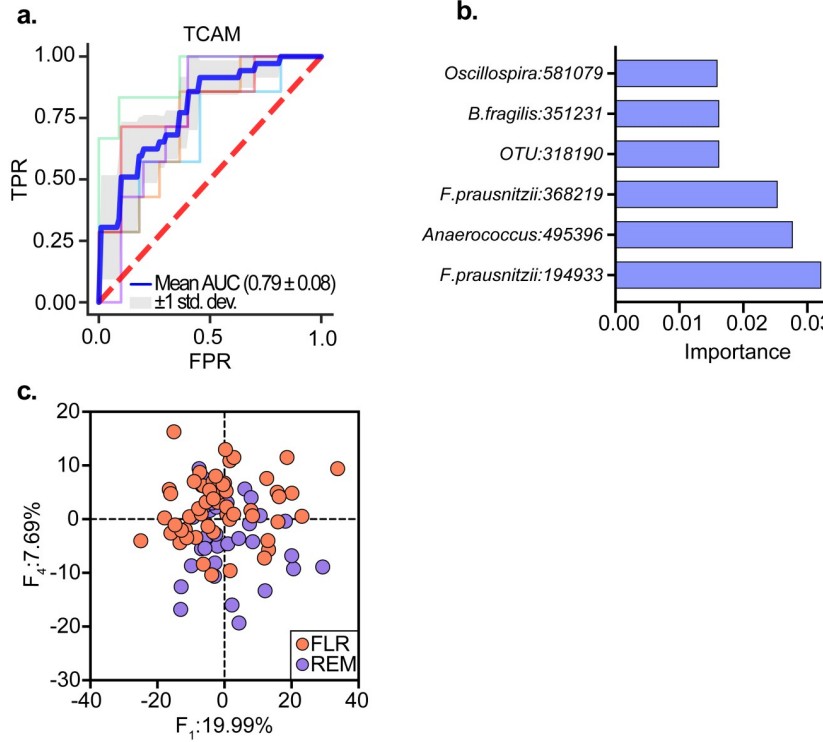

**Fig 8. TCAM enables new discoveries and is amenable for ML application. a** ROC curve for MLP model trained to classify remission/flare based on TCAM transformed data of all timepoints. **b** Bar plot showing importance scores of top 5% ranked features. **c** Scatterplot of TCAM scores computed on top 5% most important features.

We constructed an ML pipeline which, consisted of filtration, baseline normalization, TCAM based feature engineering step, followed by Multi Layer Perceptron (MLP) classifier (S1 Text).

First, an inspection of the complete dataset using TCAM, as well as PCA for the inspection of each time-point separately, failed to reveal differences between microbiome trajectories of FLR and REM groups (PERMANOVA; $P > 0.05$, S6(a), S6(b) and S6(c) Fig). This lack of clear structures in the data with respect to the labels, suggests that the task of modelling remission status using temporal microbiome data is highly challenging. Yet, remarkably, the TCAM based MLP classifier managed to achieve noticeable scores for the prediction (AUROC = 0.79, Fig 8a) when evaluated in five-fold cross-validation iterations (S1 Text). For the sake of comparison, equivalent classifiers, which take a single time-point as input (S1 Text), obtained considerably lower scores (AUROC; W12 = 0.61, W52 = 0.68, S6(d) and S6(e) Fig), similarly to the performance of a time-series classification model based on random convolutional kernel transform (Rocket [24] AUROC = 0.66, S6(f) Fig).

In addition, the straightforward out-of-sample extension of TCAM enabled downstream analysis of the classification model's decision making process. Using standard feature importance utilities, we were able to evaluate the contribution of features in the original space (OTUs) to the model's performance (Fig 8b). For example, we were able to pinpoint *Anaerococcus* and *B.fragilis* whose trajectories contribution to the decision making process were among the highest of all taxa, and additional OTUs annotated with *F.prausnitzii*, which was linked to the differences between the groups in the original paper.

To further validate the TCAM-based feature importance results, we re-applied TCAM to a reduced dataset containing features whose importance score is in the top 5%. Focusing on

these top ranked features, we were able to identify clustering according to disease states (Fig 8c), confirming that TCAM-based feature engineering truthfully preserves existing structures in the data (S6(g) Fig).

Collectively, these demonstrations highlight TCAM's ability to uncover predictive aspects underlying longitudinal data, while enabling seamless integration as a drop-in feature-engineering step in supervised ML workflows. These capacities enable to extend the transformation to samples outside the training dataset, and maintain traceable and meaningful relationships with the original features space. While it may be that other pre-processing schemes would result in better performances, especially for the Rocket based classifier, which is considered state-of-the-art, this case highlights that TCAM based classifier requires minimal pre-processing of the data in order to perform reasonably well, thus putting TCAM as a powerful method for feature engineering.

## Discussion

In this work, we present TCAM, a dimensionality reduction method for longitudinal 'omics data analysis, constructed on the premise of recent tensor-tensor algebra innovations. We demonstrate that TCAM outperforms traditional and state-of-the-art methods for longitudinal analysis dimensionality reduction, both in terms of signature detection and by pruning for meaningful features. In addition, we show that TCAM is applicable to diverse 'omics types, including amplicon and shotgun sequencing as well as proteomics. Furthermore, unlike other tensor factorization methods, TCAM entertains a natural out-of-sample extension formula, making it suitable for prediction tasks in complex experimental designs as a drop-in feature engineering utility within ML workflows. We demonstrate that we can preserve the feature importance contribution of the original features, even when TCAM is applied.

To our knowledge, TCAM is the first tensor component analysis framework that is guaranteed, within the specific choice of domain transformation, to maximize the variance of the latent representation while keeping the distortion minimal. While distinct choices of $\mathbf{M}$ would generally result in different TCAM embeddings (and transformations), the explicit rank-$q$ truncation of each resulting TCAM makes the $q$ dimensional transform maximizing the variance and minimizing distortion in the algebraic framework defined by each $\mathbf{M}$. One possible way to define the 'best' $\mathbf{M}$ for a given dataset, is the $\mathbf{M}$ for which the implicit rank of the (un-truncated) decomposition of the data is minimized. We consider this choice as the 'best' option as it surely provides the representation of the data that is the most possibly compressed. Alternatively, considering that any TSVDM (thus, any TCAM) may be written as approximation in CP form [12, Section 6.C], we get that the implicit rank under $\star_{\mathbf{M}}$ of the data equals, by definition, to its tensor-rank, making the task of finding the 'best' $\mathbf{M}$ equivalent to finding the tensor-rank of the data. Since tensor-rank computation constitutes a difficult problem in general [25], we conclude that our definition for the 'best' $\mathbf{M}$ is unhelpful as it is generally impossible to lay one's hands on. Yet, we have shown that when dealing with time-series data, taking $\mathbf{M}$ as the discrete cosine transformation, TCAM is amenable for traditional downstream applications often used in biological data analysis, such as multivariate hypothesis testing and ML workflows.

While TCAM proves to constitute a useful tool for the analysis of longitudinal experimental designs, it relies on fully sampled cohorts, i.e., where all participants provide the same number of samples corresponding to similar time points. Even though missing data imputation is a classic use-case for low-rank approximations in general [6, 9] and the recent progress made in the applications of TSVDM to incomplete data [26], the accuracy and reliability of reconstructed data generally depend on assumptions regarding the generative process of the data, the frequency of observed values or their distribution across subjects, features and timepoints.

Maintaining TCAM's universality to all kinds of 'omics data necessitates a detachment of the factorization from imputation and normalization of the data. Currently, prior to applying TCAM, it is up to the user to impute missing samples and normalize the data by any method that is appropriate to the data of interest. In this work, we have demonstrated TCAM's power in longitudinal 'omics data analysis while considering naïve and straight-forward schemes for normalization and imputation (S1 Text).

Looking forward, the mathematical properties of TCAM may enable to not only perform a trajectory analysis across time, but to also harness spatial patterns of data collected across different body sites. Future TCAM versions would enable the factorization of higher order tensors, allowing for better understanding of even more complex experimental designs, such as concomitant incorporation of space and time. A probabilistic formulation for TCAM, i.e., as a model for generating high dimensional time-series data that is subjected to some prior assumptions imposed by the type of data, may also be useful in order to handle missing data and determination of uncertainty in estimates and predictions. Moreover, the tensor-tensor **M** product framework [12], which is the theoretical foundation underlying TCAM, may be further utilized to produce additional factorization schemes, such as the decomposition of dissimilarity tensors for microbiome applications, non-negative factorizations intended for count data and more.

To conclude, the presented approach may address an important, previously unmet need for longitudinal 'omics data analysis by introducing a toolkit that enables trajectory analysis, which we make available to the wide community as a simple, one-stop-shop Python implementation (https://github.com/UriaMorP/mprod_package), that is compatible with the highly popular scikit-learn package. We believe that the application of TCAM would help derive deep insights from large-scale, longitudinal and multi-omics data, while facilitating personalized medicine-based data mining and interpretation, thereby leading to the development of tailored treatments and preventive strategies for human diseases.

## Supporting information

**S1 Text. Supplementary discussion containing proofs of main results and prepossessing details.**
(PDF)

**S1 Fig. Comparison of TCAM with existing matrix based methods for exploratory analysis. a** PCA plot of baseline timepoints, 1–2 samples per each subject. Points are colored according to participant. **b** PCA plot of all timepoints. Points are colored according to group. **c**, **d** and **e** PCA plot of baseline, antibiotics, and intervention phases respectively. Points are colored according to group.
(TIF)

**S2 Fig. Detailed view of the statistically significant features discovered using TCAM as pruning strategy (continued).** Time series of relative abundance levels for statistically significant taxa (q < 0.05, lmer), which were found using both TCAM-based pruning and without any pruning strategy **a**, or when no pruning scheme was employed **b**.
(TIF)

**S3 Fig. Detailed view of the statistically significant features discovered using TCAM as pruning strategy.** Time series of relative abundance levels for statistically significant taxa (q < 0.05, lmer), which were uniquely discovered when TCAM based pruning of the features was used.
(TIF)

**S4 Fig. Exploratory analysis of the dense ECAM dataset using TCAM. a** Scatter plot of the first two TCAM factors computed for the ECAM dataset [15]. Orange line and colored backgrounds show the boundary of decision and class domains computed using linear SVC; Inset; PERMANOVA. **b** Barplot showing the top 20 contributing features to the variation in orthogonal direction to the decision boundary. **c** Heatmap representing the cumulative change (iAUC) of the top 1% contributing features to the variation in orthogonal direction to the decision boundary; Color bar indicates z-score normalized value. **d** Time series describing relative abundances of bacteria with smallest adjusted p-value (lmer) and highest bacterial abundance after pruning strategy.
(TIF)

**S5 Fig. Application of TCAM following domain specific normalization. a** Scatterplot showing the first factors acquired by employing TCAM to the data of [16] following rclr normalization and DFB; Inset: Pairwise PERMANOVA. **b** Funnel showing the comparison between CTF (left) and TCAM + rclr (right) as pruning strategies for univariate statistical hypothesis testing. **c** Time series describing all significant bacteria (lmer) found using rclr TCAM based pruning strategy.
(TIF)

**S6 Fig. TCAM analysis on pediatric UC patients. a**, **b**, **c** Scatter plots for 2 leading factors of TCAM for the whole dataset **a** PCA computed for log2 ratio of week 12 and baseline **b** PCA computed for log2 ratio of week 52 and baseline **c** Points are colored according to remission (REM) and flare (FLR) status. **d** and **e** ROC curve for MLP model trained to classify remission/flare based on PCA transformed log fold change between week 12 and baseline (**d**), log fold change between week 52 and baseline (**e**). **f** ROC curve for ridge-regression-classifier model trained to classify remission/flare given random kernel transformation (Rocket [24]) of the complete time-series. **g** Time series of relative abundance levels, highlighting the differences in trajectories of the features contributing to the remission status classification model.
(TIF)

## Acknowledgments

E.E. is supported by the Leona M. and Harry B. Helmsley Charitable Trust, Adelis Foundation, Pearl Welinsky Merlo Scientific Progress Research Fund, Park Avenue Charitable Fund, Hanna and Dr. Ludwik Wallach Cancer Research Fund, Daniel Morris Trust, Wolfson Family Charitable Trust and Wolfson Foundation, Ben B. and Joyce E. Eisenberg Foundation, White Rose International Foundation, Estate of Malka Moskowitz, Estate of Myron H. Ackerman, Estate of Bernard Bishin for the WIS-Clalit Program, Else Kröener-Fresenius Foundation, Jeanne and Joseph Nissim Center for Life Sciences Research, A. Moussaieff, M. de Botton, Vainboim family, A. Davidoff, the V. R. Schwartz Research Fellow Chair and by grants funded by the European Research Council, Israel Science Foundation, Israel Ministry of Science and Technology, Israel Ministry of Health, Helmholtz Foundation, Garvan Institute of Medical Research, European Crohn's and Colitis Organization, Deutsch-Israelische Projektkooperation, IDSA Foundation and Wellcome Trust. E.E. is the incumbent of the Sir Marc and Lady Tania Feldmann Professorial Chair, a senior fellow of the Canadian Institute of Advanced Research and an international scholar of the Bill & Melinda Gates Foundation and Howard Hughes Medical Institute. H.A. is supported by the Israel Science Foundation, US-Israel Binational Science Foundation, and IBM Faculty Award. This research was partially supported by the Israeli Council for Higher Education (CHE) via the Weizmann Data Science Research Center.

## Author Contributions

**Conceptualization:** Uria Mor, Rafael Valdés-Mas, Haim Avron.

**Data curation:** Uria Mor, Yotam Cohen, Rafael Valdés-Mas.

**Formal analysis:** Uria Mor.

**Investigation:** Uria Mor, Yotam Cohen.

**Software:** Uria Mor, Rafael Valdés-Mas.

**Supervision:** Eran Elinav, Haim Avron.

**Validation:** Uria Mor, Yotam Cohen, Haim Avron.

**Visualization:** Denise Kviatcovsky.

**Writing – original draft:** Uria Mor, Yotam Cohen, Rafael Valdés-Mas, Denise Kviatcovsky, Eran Elinav, Haim Avron.

**Writing – review & editing:** Uria Mor, Yotam Cohen, Rafael Valdés-Mas, Denise Kviatcovsky, Eran Elinav, Haim Avron.

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
