## [Decision Letter · Decision Letter 0]

28 Mar 2022

Dear Mr. Mor,

Thank you very much for submitting your manuscript "Dimensionality Reduction of Longitudinal ’Omics Data using Modern Tensor Factorizations" for consideration at PLOS Computational Biology. As with all papers reviewed by the journal, your manuscript was reviewed by members of the editorial board and by several independent reviewers. The reviewers appreciated the attention to an important topic. Based on the reviews, we are likely to accept this manuscript for publication, providing that you modify the manuscript according to the review recommendations.

Both reviewers had some questions regarding the distinctive features and limitations of TCAM compared with some of the methods mentionned in your introduction. In particular, reviewer 1 had some concerns regarding the application of TCAM to feature engineering, which should be addressed in the revised version.

Sincerely,

Carl Herrmann, Ph.D.

Associate Editor

PLOS Computational Biology

William Noble

Deputy Editor

PLOS Computational Biology

[LINK]

Reviewer's Responses to Questions

**Comments to the Authors:**

Reviewer #1: This manuscript introduces TCAM, a tensor factorization method for the analysis of longitudinal omics data. The method is based on TSVDM, a generalized SVD for higher order tensors based on M-product framework. The authors showed that TCAM outperforms both matrix-based dimension reduction methods (i.e., PCA) and other tensor-based methods by mathematical formal analysis and real-world applications on metagenomics and proteomics data. Specifically, TCAM could capture the temporal intra-subject variations which matrix-based method could not do. Also, unlike other ALS-CP based tensor factorization methods, TCAM is proved to have optimal low rank approximation in terms of distortion and variance in the embedding space. TCAM can transform unseen data points to the embedding space and thus can be used as a feature engineering step in a ML workflow.

The manuscript is well structured and well written. While I was not able to confirm every single proof, I am confident that the topics are presented in a sound way and would make an interesting topic for the readers. The manuscript would however benefit from (1) presenting a concrete algorithm for TCAM similar to ALS algorithm for CP-based tensor factorization methods and (2) stating the computational complexity of the algorithm since higher order tensors could be suffered from the curse of dimensionality.

For the application of TCAM to supervised ML, comparing TCAM with other single time-point methods is not strong enough to support TCAM as a feature engineering step. There are extensive techniques for feature ranking and selection for ML workflows. The author could compare TCAM with those methods to show how powerful TCAM is as a feature engineering method.

One minor comment is: the citation should be in numeric order.

Reviewer #2: A novel tensor factorization method, TCAM, is proposed for analysis of longitudinal ’omics data. TCAM is derived from TSVDM, the tensor application of the popular SVD matrix factorization method. TCAM utilizes the right orthogonal tensor (V) and original data to create “scores” and “loadings” matrices describing the original data. New data can then be represented in the original data space using matrix/tensor products. In short, this method enables better evaluation of longitudinal data sets, and enables projection of new data into the existing space, enabling integration with machine learning algorithms. TCAM is demonstrated on an antibiotic intervention dataset and proteomics dataset, then described in a multilayer perceptron pipeline for a disease course dataset.

The authors do an excellent job of describing the tensor factorization comprising their TCAM method (figure 1 is well done). I was also particularly impressed with figures 2-4 describing important equations. This work adequately defends their work as a valid approach to longitudinal ’omics analysis. The use cases are well-constructed and easy to follow, but it would be interesting and potentially more convincing to see TCAM applied to denser or longer longitudinal datasets. Supplemental data is robust and well done.

Questions/Content Revisions:

What (if any) are the limitations of TCAM? Would be good to address any limitations of applying TCAM in the discussion.

How would you suggest a user determine rank ‘q’ in truncated TCAM?

Is TCAM compatible with longitudinal datasets exhibiting missed samples? In other words, do all subjects need to have all samples?

Many microbiologists use dissimilarity metrics (Bray-Curtis, Jaccard) for microbiome studies to measure sample relationships without placing too much faith in the sequencing output (known to be potentially biased). Besides centering via MDF, how else should data be prepared for TCAM to account for potentially flawed OTU counts? Is it possible to use distance/dissimilarity indices somewhere? I imagine not, since this would alter the format of the tensors (distances obviously become upper triangular matrices).

What is the distinct advantage of TCAM compared to MetaLonDA, MiRKAT, SplinectomeR, etc. mentioned in the introduction?

In the Results, did you or original authors consider looking for taxa signals at other ranks (i.e. genera, family, class)?

I believe answering some or all of these questions could strengthen the manuscript, at the discretion of the authors.

Minor Revisions:

Methods: Short paragraph describing MDF abruptly transitions to TSVDM then returns to mentioning MDF in TCAM part. Could rearrange or make this paragraph connect to TSVDM better.

Results: (spelling correction) “Mutlivariate” in 2nd paragraph

Results: (spelling correction) “non of” in 4th paragraph

**Have the authors made all data and (if applicable) computational code underlying the findings in their manuscript fully available?**

Reviewer #1: Yes

Reviewer #2: Yes

PLOS authors have the option to publish the peer review history of their article (what does this mean?). If published, this will include your full peer review and any attached files.

Reviewer #1: No

Reviewer #2: No

Figure Files:

Data Requirements:

Reproducibility:

References:

---

## [Decision Letter · Decision Letter 1]

16 May 2022

Dear Mr. Mor,

We are pleased to inform you that your manuscript 'Dimensionality Reduction of Longitudinal ’Omics Data using Modern Tensor Factorizations' has been provisionally accepted for publication in PLOS Computational Biology.

Best regards,

Carl Herrmann, Ph.D.

Associate Editor

PLOS Computational Biology

William Noble

Deputy Editor

PLOS Computational Biology

Reviewer's Responses to Questions

**Comments to the Authors:**

Reviewer #1: Thanks the authors for addressing my concerns. I do not have further comments.

Reviewer #2: Thank you to all authors for their thorough consideration of my questions and comments. Everything in my original review has now been addressed in the proposed manuscript, and I believe this innovative, well-supported work will be well-received in the field upon publication.

**Have the authors made all data and (if applicable) computational code underlying the findings in their manuscript fully available?**

Reviewer #1: None

Reviewer #2: Yes

PLOS authors have the option to publish the peer review history of their article (what does this mean?). If published, this will include your full peer review and any attached files.

Reviewer #1: No

Reviewer #2: No

---

## [Editor Report · Acceptance letter]

8 Jul 2022

PCOMPBIOL-D-22-00201R1 

Dimensionality Reduction of Longitudinal ’Omics Data using Modern Tensor Factorizations

Dear Dr Mor,

I am pleased to inform you that your manuscript has been formally accepted for publication in PLOS Computational Biology. Your manuscript is now with our production department and you will be notified of the publication date in due course.

With kind regards,

Zsofi Zombor
